# Prevalence of Multidrug-Resistant *Salmonella enterica* Serovars in Buffalo Meat in Egypt

**DOI:** 10.3390/foods11182924

**Published:** 2022-09-19

**Authors:** Samir Mohammed Abd-Elghany, Takwa Mohammed Fathy, Amira Ibrahim Zakaria, Kálmán Imre, Adriana Morar, Viorel Herman, Raul Pașcalău, Laura Șmuleac, Doru Morar, Mirela Imre, Khalid Ibrahim Sallam

**Affiliations:** 1Department of Food Hygiene and Control, Faculty of Veterinary Medicine, Mansoura University, Mansoura 35516, Egypt; 2Department of Animal Production and Veterinary Public Health, Faculty of Veterinary Medicine, Banat’s University of Agricultural Sciences and Veterinary Medicine “King Michael I of Romania”, 300645 Timişoara, Romania; 3Department of Infectious Diseases and Preventive Medicine, Faculty of Veterinary Medicine, Banat’s University of Agricultural Sciences and Veterinary Medicine “King Michael I of Romania”, 300645 Timişoara, Romania; 4Department of Agricultural Technologies, Faculty of Agriculture, Banat’s University of Agricultural Sciences and Veterinary Medicine “King Michael I of Romania”, 300645 Timişoara, Romania; 5Department of Sustainable Development and Environmental Engineering, Faculty of Agriculture, Banat University of Agricultural Sciences and Veterinary Medicine “King Michael I of Romania”, 300645 Timişoara, Romania; 6Department of Internal Medicine, Faculty of Veterinary Medicine, Banat’s University of Agricultural Sciences and Veterinary Medicine “King Michael I of Romania”, 300645 Timişoara, Romania; 7Department of Parasitology and Parasitic Diseases, Faculty of Veterinary Medicine, Banat’s University of Agricultural Sciences and Veterinary Medicine “King Michael I of Romania”, 300645 Timişoara, Romania

**Keywords:** *Salmonella* serovars, antimicrobial resistance, buffalo meat, PCR, virulence genes

## Abstract

The current study aimed to investigate the presence of *Salmonella* spp. prevalence in buffalo meat in Egypt, along with studying the antimicrobial susceptibility of the recovered isolates. *Salmonella* spp. was detected in 25% of tested buffalo meat. A total of 53 (100%) isolates were genetically verified by PCR as *Salmonella*, based on the detection of the *invA* gene. The *stn* and *hilA* genes were detected in 71.7% (38/53), and 83.0% (44/53) of the recovered isolates, respectively. *Salmonella* Enteritidis (11/53; 20.7%) was the most commonly isolated serovar, followed by *S*. Typhimurium (9/53; 17%), *S*. Montevideo (6/53; 11.3%), meanwhile, *S*. Chester, *S*. Derby, *S*. Papuana, and *S*. Saintpaul were the least commonly identified serovars (a single strain for each; 1.9%). Among the 16 antimicrobials tested, amikacin, imipenem, gentamicin, cefotaxime, meropenem, ciprofloxacin, and enrofloxacin were the most effective drugs, with bacterial susceptibility percentages of 98.1%, 94.3%, 92.5%, 86.8%, 83.0%, 73.6%, and 69.8%, respectively. Meanwhile, the least effective ones were erythromycin, streptomycin, clindamycin, cefepime, and nalidixic acid, with bacterial resistance percentages of 100%, 98.1%, 88.7%, 77.4%, and 66%, respectively. Interestingly, the high contamination level of Egyptian buffalo meat with multidrug-resistant *Salmonella* (79.2%; 42/53) can constitute a problem for public health. Therefore, programs to control *Salmonella* contamination are needed in Egypt.

## 1. Introduction

Buffalo is a good source of meat, especially in areas where the climate adversely affects the production efficiency of other animals. The demand for buffalo meat appears to be increasing, especially in arid regions because of its unique physiological characteristics including strong musculature and genetic characteristics enabling them to be more disease resistant than other animals and can tolerate a broad range of environmental and nutritional changes as high temperatures, solar radiation, water scarcity, rough topography and poor vegetation [1].

Buffalo meat constitutes the largest proportion of the native red meat produced among the red meat-producing animals in Egypt. The Egyptian Ministry of Agriculture and Land Reclamation (MALR) revealed that the large live animals producing red meat in Egypt during the market year 2019 consisted of 4.9 million cows and 3.48 million buffalo. These produced a calf crop of 1.134 and 1.089 million heads of cow and buffalo calves slaughtered in the governmental slaughterhouses to yield 366,000 and 381,000 tons of beef and buffalo meat, which constituted 43.5% and 45.3% of the native red meat produced in the country, respectively.

Buffalo meat is also an important source of the healthiest red meat, due to its nutritive value, as it contains higher protein, iron content, and polyunsaturated fatty acid with low cholesterol content needed for consumer health, in comparison to other meat animals. Furthermore, buffalo meat has superior meat processing characteristics, due to its chemical composition, structural components, and functional abilities. The major gorgeous features of buffalo meat include its dark red color, good marbling, little connective tissue, desirable texture, high protein content, water holding capacity, myofibrillar fragmentation index value, and emulsifying capacity, so it enters in the manufacturing of many meat products [2].

The meat and its derived products are rich in water and proteins with high biological values, which makes it an essential food for a balanced diet. However, these properties make it a favorable medium for microbial growth. The carcasses of the healthy animals are almost sterile, with the contamination mostly occurring after different stages of the slaughtering process from the dirty skin, hooves, ruminal and intestinal contents, knives, polluted air and water, infected personnel, faulty slaughtering procedures, handling during processing, mincing machine and storage [3,4]. The contamination might be caused by bacteria that are either spoilage bacteria that cause meat putrefaction or pathogenic bacteria that are implicated in foodborne illness. The most frequent foodborne pathogens that have been implicated with meat and its products comprise *Salmonella* spp., *Campylobacter* spp., *Staphylococcus aureus*, and *Escherichia coli* O157:H7, which pose a probable danger to the consumer’s health [3,5,6].

At present, *Salmonella enterica* serovars are reported worldwide as the most common cause of human gastroenteritis associated with the consumption of contaminated foods. *S. enterica* is a very important food-borne bacteria that results in several public health troubles for humans and animals [7]. It is considered to be the second bacterial foodborne disease infecting humans. Worldwide, it results in approximately 93.8 million cases of acute gastroenteritis, along with 155,000 human deaths, and most human outbreaks (nearly 80 million cases) are associated with the ingestion of contaminated food with non-typhoidal *Salmonella* [8]. Although *Salmonella* spp. holds an uppermost position among the foodborne pathogens in Egypt, there is no national surveillance of reliable statistical data concerning the illness and economic dilemma of foodborne salmonellosis in the country [9]. The supreme predominant serovars recovered from foodborne infection cases were *Salmonella* Typhimurium and *Salmonella* Enteritidis, which account for 46%, and 24% of gastroenteritis outbreaks, respectively [10]. The symptoms of human gastroenteritis caused by *Salmonella* may include diarrhea, fever, and abdominal cramps with an incubation period ranging from 12 h to 3 days. The illness generally lasts 4 to 7 days and the majority of infected cases recover without medication. The infective dose of *Salmonella* spp. could be very low of about 15–20 bacterial cells, depending on the human age, the health condition of the host, and the implicated *Salmonella* serovars. Older people, young children, and immunosuppressed persons are more susceptible to severe symptoms [11,12]. The incidence of human salmonellosis cases with more difficulty in the treatment of such infection is increased in the last decades due to the emergence of multidrug-resistant (MDR) *Salmonella* strains, particularly *S*. Typhimurium and *S*. Enteritidis isolated from different animal origin foods worldwide [13,14]. The increased prevalence rates of antimicrobial-resistant microorganisms during the course of food production is possible due to the extensive use of the common antimicrobials (particularly overuse or misuse) as growth promoters, prophylactic, or therapeutics, in food-producing animals [15]. The emergence of MDR *Salmonella* has become a global public health problem, as it gravitates toward more virulence [9].

To date, limited information is available concerning the *Salmonella* spp. isolation from Egyptian buffalo meat [16,17]. Therefore, considering the increased marketing and consumption of buffalo meat in Egypt, and the global increase in the prevalence of MDR *Salmonella* isolates against the most ordinarily used antimicrobials, this work was planned to study the prevalence, serotyping, virulence genes existence and the phenotypic antimicrobial resistance profile of *Salmonella* isolates from buffalo meat distributed in Mansoura, Egypt.

## 2. Materials and Methods

### 2.1. Sample Collection

A total of 100 buffalo meat samples (each sample represented by 250 g) were collected from the retail butcher shops located in Mansoura city, Egypt, from November 2020 to June 2021. Each sample was individually wrapped into a sterile bag, labeled, and then transferred rapidly in a refrigerated box at ∼4 °C to the Laboratory of Food Hygiene and Control Department, from the Faculty of Veterinary Medicine, Mansoura University, Egypt, for bacteriological analyses. The specified mass of the test meat portion used for isolation of *Salmonella* spp. was represented by a quantity of 25 g, which was sampled aseptically, using a sterile scalpel, from each tested sample.

### 2.2. Isolation and Identification of Salmonella spp.

The detection of *Salmonella* spp. was performed according to ISO [18]. In brief, 25 g meat portion from each buffalo meat sample were aseptically excised and introduced into a sterile homogenizer flask containing 225 mL sterile buffered peptone water (CM0509B; Oxoid Ltd., Basingstoke, UK), and thoroughly stirred for one min, in a stomacher (Stomacher 400 Lab Blender; Seward Medical, London, UK). Each mixed meat sample was then incubated at 37 °C for 24 h. From pre-enriched culture, volumes of 0.1 mL and 1 mL, respectively were aseptically inoculated into 10 mL each of Rappaport–Vassiliadis *Salmonella* enrichment broth (RV; CM0669; Oxoid Ltd., Basingstoke, UK) and Muller–Kauffmann tetrathionate/novobiocin broth (MKTTn; CM1048; Oxoid Ltd., Basingstoke, UK), respectively, followed by incubation at 42 °C for 24 h, and 37 °C for 24 h, consecutively. A loopful from RV and MKTTn broths was streaked onto two selective solid media; Xylose-Lysine-Desoxycholate (XLD) agar (CM0469; Oxoid Ltd., Basingstoke, UK) and Brilliant Green Agar w/Sulfadiazine (BGA; NC7310; Neogen Ltd., Maharashtra, India) plates, which were incubated for 24 h at 37 °C, and at 35 °C, respectively. Approximately five typical suspected colonies of presumptive *Salmonella* (pink color, with or without black centers), were picked up and purified onto nutrient agar slopes, then incubated for 24 h at 37 °C, for biochemical and serological confirmation.

After the application of Gram staining and motility tests, the biochemical tests for the presumed *Salmonella* isolates were conducted based on the production of indole from tryptophan (Tryptophan broth, MerckKGaA, Darmstadt, Germany), hydrogen sulfide, gas, and sugar fermentation (TSI test; CM0277, Oxoid Ltd., Basingstoke, UK), methyl red (MRVP medium; CM0043, Oxoid Ltd., Basingstoke, UK), Voges–Proskauer (MRVP medium; CM0043, Oxoid Ltd., Basingstoke, UK), urease (Urea broth base, CM0071, Oxoid Ltd., Basingstoke, UK), citrate utilization (Simmons citrate agar, CM0155, Oxoid Ltd., Basingstoke, UK), and oxidase (oxidase detection strips, MB0266, Oxoid Ltd., Basingstoke, UK) tests, in addition to carbohydrate, glucose and lactose fermentation, and lysine decarboxylation (Lysin iron agar, CM0381, Oxoid Ltd., Basingstoke, UK). The biochemically confirmed isolates of presumptive *Salmonella* spp. detected by the standardized culture methods in the present study were further verified by the polymerase chain reaction (PCR) assay.

As has been previously indicated by several authors [3,9,10,12,16,17], it is important to mention that more than one *Salmonella* isolate was tested from a single positive meat sample for the following reasons: (*i*) meat samples can contain more than one *Salmonella* serovars, (*ii*) two or more of the same *Salmonella* serovars can exhibit different antimicrobial resistant patterns, (*iii*) more than one isolates of the same or different *Salmonella* serovar taken from a single positive sample can exhibit different existence profiles of the virulence genes tested.

### 2.3. Molecular Analysis

The chromosomal DNA was isolated according to Abd-Elghany and Sallam [19], using the QIAamp DNA Mini Purification kit (Qiagen, Germany), and following the producer’s recommendations.

Genomic DNA of *Salmonella* Typhimurium (RIMD 1985009) and *Escherichia coli* K12DH5α acquired from National Research Centre (NRC), Dokki, Cairo, Egypt was served as positive and negative control reference strains, respectively, for the presence and absence of *invA*, *stn*, and *hilA* genes. Phenotypically-identified *Salmonella* isolates, detected from examined buffalo meat samples, were molecularly identified by using the multiplex polymerase chain reaction assays, based on the presence of *invA* marker gene targeting for invasion of *Salmonella* in the host organism, using specific oligonucleotide primer sequences constructed to amplify DNA amplicon of 275 bp for the *invA* gene [20]. The identified strains were then tested for the detection of *stn* and *hilA* virulence genes, using specific oligonucleotide primer sequences designed for PCR amplification of DNA fragments at a molecular size of 617 bp for the *stn gene* [21] and 854 bp for the *hilA* gene [22].

Multiplex PCR was carried out using a thermal cycler (Eppendorf Mastercycler, Hamburg, Germany), in a 50-μL volume, which includes 1 μL *Salmonella* genomic DNA template, 2 μL each of forward and reverse primers (10 pmol each), 25 μL DreamTaq Green Master Mix (Thermo Scientific, St. Leon Roth, Germany), and 20 μL of sterile distilled water. The denaturation step at 94 °C for 4 min, 35 cycles (94 °C, for 45 s, 56 °C, for 60 s, 72 °C, for 60 s) was followed by a final extension, at 72 °C, for 7 min. The PCR-amplified products of each reaction mixture were separated by subjecting 4 µL aliquots to agarose (1.5%) gel electrophoresis (Cleaver Scientific–Horizontal Electrophoresis System; Cleaver Scientific Ltd., Rugby, UK), for 30 min, at 100 V, followed by 25-min staining in 1% solution of ethidium bromide solution. Finally, the separated PCR products were visualized and photographed under UV illumination. Amplified genes were checked by DNA sequencing with the BigDye Terminator version 3.1 Cycle Sequencing Kit, following the producer’s recommendations on an ABI Prism 3100 automated sequencer (Applied Biosystems), using the same primers as those used for gene amplification.

PCR-verified *Salmonella* isolates were then subjected to slide and tube agglutination technique for *Salmonella* serotyping based on the detection of somatic (O), and flagellar (H) antigens by using separated O and H *Salmonella* antisera (Denka Seiken Co., Tokyo, Japan) according to the White-Kauffmann-Le Minor scheme.

### 2.4. Antimicrobial Susceptibility Tests

The antimicrobial susceptibility of the *Salmonella* strains was evaluated using the agar disk diffusion standard method according to the Clinical and Laboratory Standards Institute guidelines [23]. Antimicrobial susceptibility discs (Difco Laboratories, and BioMerieux, France) of different concentrations were used to detect susceptibility or resistance of the isolated *Salmonella* strains on Mueller–Hinton Agar (Oxoid Ltd., Basingstoke, UK), at the following drug concentrations: amikacin (AMK; 30 μg), ampicillin (AMP; 10 μg), cefepime (CPM; 30 μg), cefotaxime (CTX; 30 μg), cefalotin (CET; 30 μg), ciprofloxacin (CIP; 5 μg), clindamycin (CM; 2 μg), enrofloxacin (ENR; 5 μg), erythromycin (E; 15 μg), gentamicin (GEN; 10 μg), imipenem (IPM; 10 μg), meropenem (MEM; 10 μg), nalidixic acid (NA; 30 μg), streptomycin (S; 10 μg), trimethoprim/sulfamethoxazole (SXT; 25 μg), tetracycline (TE; 30 μg). Strains were evaluated as susceptible, intermediate resistant, or resistant, according to Clinical and Laboratory Standards Institute guidelines [23]. *Escherichia coli* ATCC 25922 was used as a positive control strain. Multiple antibiotic resistances (MARs) index for each resistance pattern was calculated using the formula MAR Index = Number of resistance antimicrobials/total number of antimicrobials tested; where a MAR index > 0.2 reveals high-risk contamination [24].

## 3. Results and Discussion

### 3.1. Prevalence of the Isolated Salmonella enterica Serovars in Buffalo Meat

The current study indicated that 25 (25%) of the 100 buffalo meat samples monitored were contaminated with *Salmonella* spp. based on the molecular confirmation of the existence of the *Salmonella* marker gene; the *invA* gene by the PCR technique, which was conducted on the genome of the presumptive *Salmonella* isolates.

Variable detection rates of *Salmonella* spp. in buffalo meat had been obtained by several research groups worldwide, using conventional microbiological methods. Our results are in close agreement with those published by Mezali and Hamdi [25], in Algeria, who could isolate *Salmonella* spp. from 23.61% of raw red meat and different meat products. Higher incidences of *Salmonella* spp., of 80%, and 46.67%, however, were recorded in buffalo meat in Laos [26], and Bangladesh [27], respectively. Conversely, lower incidences of 7% [28], 7.11% [29], 18% [30], 10.66% [31], and 7.4% [32] were reported for *Salmonella* in buffalo meat samples from Iran, Laos, Egypt, India, and Nepal, respectively.

The broad variation in *Salmonella* prevalence in the examined samples from different investigations could be related to several factors including the seasonal effects, geographical provenience, the used sampling, and bacteriological techniques, as well as the degree of the slaughter hygiene and the occurrence of products cross-contamination during different stages of buffalo dressing and preparation [33].

### 3.2. Distribution of Salmonella serovars Isolated from Buffalo Meat Samples

There were 14 different *Salmonella* serovars identified among the isolates (Figure 1). Out of them, *Salmonella* Enteritidis (11/53; 20.7%) was the most frequently encountered, followed by *S*. Typhimurium (9/53; 17%), *S*. Montevideo (6/53; 11.3%), meanwhile, *S*. Derby, *S*. Saintpaul, *S*. Papuana, and *S*. Chester were the least commonly identified serovars (a single strain for each 1.9%) (Figure 1).

The dominant occurrence of *S*. Enteritidis and *S*. Typhimurium serovars in the present investigation was also pointed out by the results of other several studies of foodborne salmonellosis outbreaks, in which *S*. Enteriditis and *S*. Typhimurium were isolated at high rates from the examined buffalo meat samples, from different countries [28,34,35,36]. In agreement with the results of some previously conducted investigations [16,17], the current study strengthens the fact that Egyptian buffalo meat could be considered a major potential source of human salmonellosis.

### 3.3. PCR Confirmation of the Salmonella Isolates

Multiplex PCR (Figure 2) method was carried out to target the conserved regions of *Salmonella* spp., such as the *invA*, *stn*, and *hilA* genes [9,37,38]. PCR amplification results revealed that only 53 (65.4%) out of the 81 biochemically identified isolates from the examined buffalo meat specimens were confirmed as *Salmonella* spp. by the *invA*-targeted gene. The present study indicated that all (100%; 53/53) serologically identified *Salmonella* isolates (n = 53) examined were molecularly confirmed by multiplex PCR to be contaminated with *Salmonella* spp., based on the presence of the *invA* gene (275 bp), meanwhile *stn* gene (617 bp), and *hilA* gene (854 bp) (Figure 2) were detected in 71.7%, and 83% of *Salmonella* recovered isolates, respectively (Figure 3).

The distribution of the virulence genes detected among the different *Salmonella* serovars identified is fully described in Table 1.

The *invA* gene is one of the most widely used genetic markers for the detection of *Salmonella* spp. in food origin samples, which is responsible for invasion in the host cells [39]. Its presence is strongly associated with the occurrence of other virulence genes, such as the *stn* gene, with important contributions to the pathogenicity process, that causes an enterotoxin effect on host cells, leading to gastroenteritis with diarrhea [40], and the *hilA* gene, which participated in the adhesion and invasion processes of the pathogen to the host cells, and increases *Salmonella* pathogenicity [39]. The presence of these genes aids the organisms to interact with the host cells and may indicate the virulence potential of *Salmonella* spp. The PCR product specificity was evaluated by sequencing the amplified *invA*, *stn*, and *hilA* fragments. The sequenced amplicons had 100% similarity with the analogous regions of *invA*, *stn*, and *hilA* genes available in the GenBank^®^.

Our finding of the *invA* gene among *Salmonella* serovars was nearly similar to those obtained by Sallam et al. [40], in Egypt, Thung et al. [41], in Malaysia, and Dong et al. [42], in China, who reported that all (100%) of the tested *Salmonella* isolates were positive for *invA* gene-specific for *Salmonella* spp. Interestingly, Yanestria et al. (2019) reported the detection of the *invA* gene in only 12.5% of the tested milkfish-origin *Salmonella* isolates [43]. In the same context, the high identification rate of the *hilA* gene (78.5%) among the tested *Salmonella* isolates in the present study was compatible with that reported by Thung et al. [41], who reported the presence of *hilA* gene in 82.6% of recovered *Salmonella* isolates from beef samples in Malaysia. Meanwhile, the detection rate of the *stn* gene in our *Salmonella* isolates (71.4%) was lower than that recorded in a previously conducted investigation by Sallam et al. [40], who detected the *stn* gene in all (100%) examined *Salmonella* isolates from beef samples in Egypt.

Results of previously conducted studies suggested that the PCR technique, targeting the amplification of the *invA* gene, showed better specificity, sensitivity, and accuracy compared with conventional methods (e.g., cultural on XLD plates and biochemical methods), besides its capacity in identifying *Salmonella* isolates from buffalo meat specimens within few hours. Therefore, those characteristics make PCR techniques a suitable and complementary method for confirmation and identification of *Salmonella* isolates from the examined samples [39,40].

### 3.4. Antimicrobial Resistance of Salmonella Isolates

The results of the disc diffusion test conducted for antibiotic resistance of buffalo meat origin *Salmonella* isolates indicated that out of 16 tested antimicrobials, the most effective drugs were AMK, IPM, GEN, CTX, MEM, CIP, ENR, TE, and AMP, which exhibited bacterial sensitivity percentages of 98.1%, 94.3%, 92.5%, 86.8%, 83.0%, 73.6%, 69.8%, 58.5%, and 56.6%, respectively. The least effective antimicrobials against the tested *Salmonella* isolates were E, S, and CM, with bacterial resistance percentages of 100%, 98.1%, and 94.3%, respectively (Table 2). The high resistance rates of *Salmonella* isolates against the aforementioned antimicrobials, which are widely used in human as well as veterinary medicine in Egypt are probably due to their over-usage as therapeutics, prophylactic agents, or growth promoters in the livestock veterinary medicine of the screened region [9,16,44,45].

Although restrictive measures are applied to reduce the prevalence of *Salmonella* Typhimurium and *S*. Enteritidis in different countries, they are not healthcare providers for reportable diseases in Egypt. Unfortunately, *Salmonella* infections are therapeutically treated in Egypt, and such a way is not sufficient to clean hosts from *Salmonella* spp., which can remain in live stocks, often in latent form. The highest resistance rate of *Salmonella* isolates against E (100%), and NA (66.0%) in our study is in accordance with that obtained by Sallam et al. [40], in Egypt, who found a high resistance percentage by *Salmonella* isolates against E (100%), and NA (70%). These results are also in accordance with those reported by Hassan et al. [25], who found that all buffalo meat origin *Salmonella* isolates were susceptible to CIP and GEN, in Bangladesh. On the other hand, the low rate of resistance against AMK, IPM, GEN, CTX, MEM, CIP, and ENR in the present investigation could be related to their non-use or low-frequency use in animal production; hence these antimicrobials are considered the drug of choice in the management of human *Salmonella* infections. The high susceptibility played by the *Salmonella* strains against CIP and ENR was not surprising, since the antimicrobial resistance toward these molecules is reported worldwide because of their large use in food-producing animals [36,44,45].

### 3.5. Antimicrobial Resistance Profiles of Salmonella Isolates

The expressed AMR profiles and MAR indexes of the tested isolates are summarily presented in Table 3. The 53 resistant isolates comprised 16 resistance profiles. The E + S + CM + CPM (*n* = 6), E + S + CM (*n* = 6), E + S + CM + CPM + NA (*n* = 6) profiles were the most frequently encountered, followed by E + S + CM + CPM + NA + SXT + CET + AMP + TE + ENR (*n* = 5), E + S (*n* = 5), E + S + CM + CPM + NA + SXT + CET (*n* = 5), E + S + CM + CPM + NA + SXT + CET + AMP + TE + ENR + CIP (*n* = 4) and E + S + CL + FEP + NA + SXT + CN + AMP (*n* = 4). The great majority (77.3%, 41/53) of the isolates showed resistance to four, or more tested antimicrobials, from different classes. Accordingly, isolates expressing resistance to at least one drug, in a minimum of three or more antimicrobial classes, were considered multidrug-resistant [44], in which the MAR index was > 0.2. These MDR strains comprised all *S*. Enteritidis (*n* = 11), *S*. Infantis (*n* = 5), *S*. Essen (*n* = 3), *S*. Tsevie (*n* = 2), and *S*. Dublin (*n* = 2) strains. In addition, seven strains of *S*. Typhimurium, five strains of *S*. Montevideo, five strains of *S*. Rissen, three strains of *S*. Virchow, and one strain each of *S*. Anatum, *S*. Derby, *S*. Papuana, and *S*. Saintpaul were also MDR (Table 4). Only six (11.3%) out of the 53 isolates expressed resistance to just one or two antimicrobials, and these included two strains of *S*. Chester and one strain each of *S*. Typhimurium, *S*. Montevideo, *S*. Virchow, and *S*. Anatum, in which MAR index was 0.125 or less, indicating that there was no MDR (Table 3). A MAR index value higher than 0.2 is considered high risk, while a value lower than 0.2 indicates low risk [45].

In the current investigation, the occurrence of MDR strains (77.3%) is almost similar to that published by Boonmar et al. [26], who stated that 73% of *Salmonella* isolates (44/60) recovered from meat samples in Laos showed MDR. Higher MDR (100%) was observed by Sallam et al. [40], in Egypt, while a lower incidence (32.2%) of MDR *Salmonella* isolates was recorded in Algeria by Mezali and Hamdi [25]. The appearance of MDR among *Salmonella* isolates is a concern for public health, which needs more caution to prevent the hazardous and unnecessary use of antimicrobials in food industries and veterinary fields. The differences in the resistance patterns found in different investigations might be due to variations between the sampled geographical areas, locally used drugs, farm management practices, and misuse or overuse of some antimicrobials.

### 3.6. Categorization of Salmonella Isolates Based on Their Antimicrobial Resistance Profiles

In the current study, 1.9%, 7.5%, 11.3%, and 79.2% of *Salmonella* isolates (*n* = 53) were categorized, according to their resistance levels against the 16 tested commonly used antimicrobials, into pan-drug-resistant (PDR), extensively drug-resistant (XDR), low-drug resistant (LDR), and multi-drug resistant (MDR), respectively (Table 4), according to the description of Magiorakos et al. [44].

Interestingly, the only one isolate that showed a pandrug-resistant pattern (resistant to all of the 16 antimicrobials tested) was *S*. Enteritidis, while the rest of *S*. Enteritidis serovar isolates (n = 9) besides seven (87.5%) of the eight *S*. Typhimurium identified in the present study were multidrug-resistant (resistant to more than 10 of the antimicrobials tested). Additionally, the four isolates, which showed an extensively drug-resistant pattern (resistant to 13 or more of the antimicrobials tested) were distributed as one from each of *S*. Enteritidis, *S*. Typhimurium, *S*. Montevideo, and *S*. Rissen (Table 3 and Table 4). On the other, hand all of the identified *S*. Dublin, *S*. Anatum, *S*. Derby, *S*. Papuana, *S*. Saintpaul, and *S*. Chester were resistant to 10 or less of the antimicrobials tested. Our findings concerning the antimicrobial resistance pattern of *Salmonella* serovars, especially the predominant ones (*S*. Enteritidis and *S*. Typhimurium) substantiate what had been reported in various studies originated from different countries [46,47,48,49].

*Salmonella* strains isolated in the present survey indicated 16 different AMR patterns, with an average MARs index of 0.436, and 77.3% (41/53) of them showed MAR indices > 0.2. A value of MAR index > 0.2 indicated an overuse and/or misuse of antibiotics. The current results for the MAR index reveal higher resistance rates among *Salmonella* isolates. Therefore, establishing valuable national surveillance systems in order to monitor the rational use of antimicrobials in the field of veterinary medicine is crucial. Likewise, for a better understanding of the antimicrobial susceptibility pattern of the isolated *Salmonella* strains, further studies are recommended focusing on the monitoring of their genotypic resistance pattern.

## 4. Conclusions

This study concluded that a high percentage of buffalo meat, marketed in Mansoura City, Egypt, was contaminated with *Salmonella* spp., with a dominant occurrence of *S*. Enteritidis (11/53) and *S*. Typhimurium (9/53) serovars. The great majority of *Salmonella* (~89%, 47/53) isolates were multidrug-resistant with an average MAR index of 0.436, and 77.3% (41/53) of them showed MAR indices > 0.2. which indicated an overuse and/or misuse of the antibiotics. Additionally, the virulence of *Salmonella* isolates was determined by the existence of the *invA*, *stn*, and *hilA* genes, which were detected in 100%, 71.7%, and 83% of *Salmonella* recovered isolates, respectively. Therefore, buffalo meat can constitute a significant public health concern, emphasizing the necessity of the implementation of a better antimicrobial stewardship program in Egypt, in order to decrease the unnecessary use of antimicrobials in food-producing animals. In addition, continuous efforts are required to maintain the disease prevalence low, applying control measures based on the prevention of contamination of animal meat at slaughter and butcher shops level, as well as proper cooking of meat at private households.

## Figures and Tables

**Figure 1 foods-11-02924-f001:**
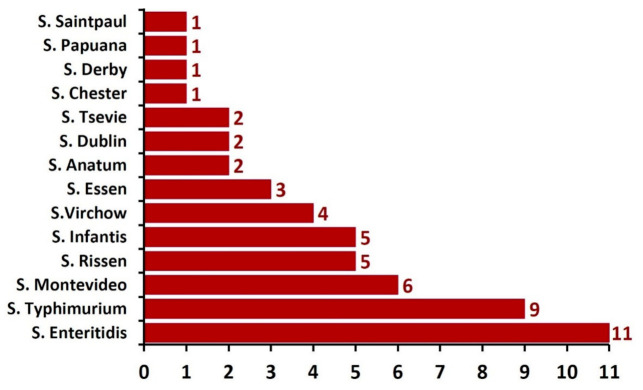
Distribution of the identified *Salmonella* serovars (n = 53) isolated from buffalo meat samples.

**Figure 2 foods-11-02924-f002:**
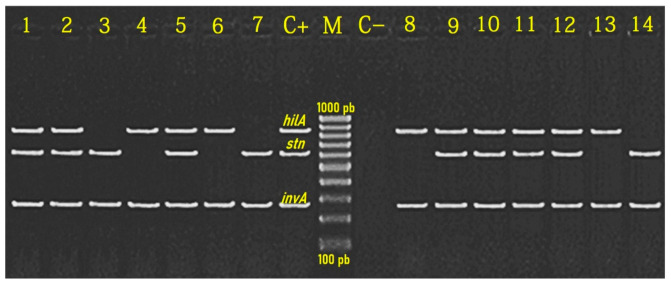
Agarose gel electrophoresis of multiplex PCR amplicons of *invA* (275 bp), *stn* (617 bp), and *hilA* (854 bp) virulence genes of *Salmonella* spp. Lane M: 100-bp ladder as a molecular size DNA marker (Jena Bioscience GmbH). Lane C+: Control positive strain for *invA*, *stn*, and *hilA* genes. Lane C–: Control negative. Lanes 1 (S. Enteritidis), 2 (*S*. Typhimurium), 5 (*S*. Infantis), 9 (*S*. Dublin), 10 (*S*. Anatum), 11 (*S*. Derby), and 12 (*S*. Papuana) showed positive bands for *invA*, *stn* and *hilA* genes. Lanes 3 (S. Montevideo), 7 (*S*. Essen), and 14 (*S*. Chester) showed positive bands for *invA* and *stn* genes. Lanes 4 (*S*. Rissen), 6 (*S*. Virchow), 8 (*S*. Tsevie), and 13 (*S*. Saintpaul) showed positive bands for *invA* and *hilA* genes.

**Figure 3 foods-11-02924-f003:**
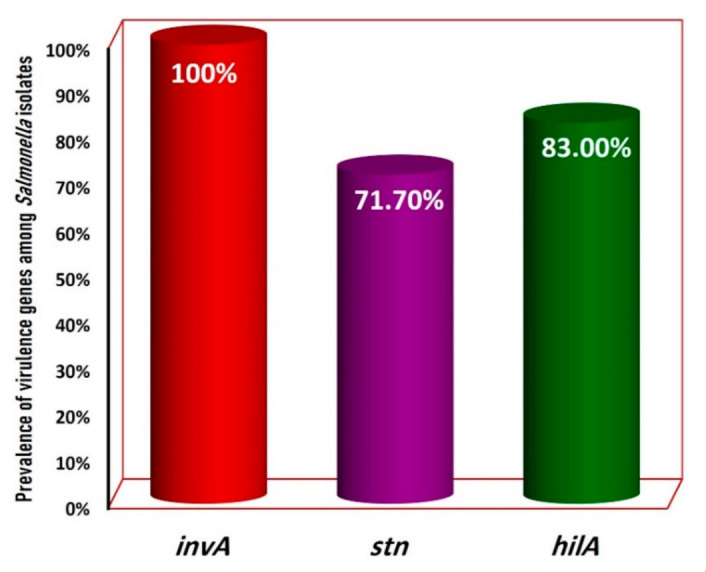
Prevalence of virulence genes among the screened *Salmonella* isolates (n = 53).

**Table 1 foods-11-02924-t001:** Prevalence of the investigated virulence genes among the identified *Salmonella* serovars.

Serovars (Number)	Positivity of the *Salmonella* Serovars for the Tested Virulence Genes		
	*invA*	*stn*	*hilA*
*S*. Enteritidis (9)	**+**	**+**	**+**
*S*. Enteritidis (1)	**+**	**−**	**+**
*S*. Enteritidis (1)	**+**	**+**	**−**
*S*. Typhimurium (7)	**+**	**+**	**+**
*S*. Typhimurium (1)	**+**	**−**	**+**
*S*. Typhimurium (1)	**+**	**+**	**−**
*S*. Montevideo (3)	**+**	**+**	**−**
*S*. Montevideo (3)	**+**	**+**	**+**
*S*. Rissen (5)	**+**	**−**	**+**
*S*. Infantis (4)	**+**	**+**	**+**
*S*. Infantis (1)	**+**	**−**	**+**
*S*. Virchow (3)	**+**	**−**	**+**
*S*. Virchow (1)	**+**	**+**	**−**
*S*. Essen (2)	**+**	**+**	**−**
*S*. Essen (1)	**+**	**+**	**+**
*S*. Anatum (2)	**+**	**−**	**+**
*S*. Dublin (2)	**+**	**+**	**+**
*S*. Tsevie (1)	**+**	**+**	**+**
*S*. Tsevie (1)	**+**	**−**	**+**
*S*. Chester (1)	**+**	**+**	**+**
*S*. Derby (1)	**+**	**+**	**+**
*S*. Papuana (1)	**+**	**−**	**+**
*S*. Saintpaul (1)	**+**	**+**	**−**
**Total (53)**	**53 (100%)**	**38 (71.7%)**	**44 (83.0%)**

Legend: *invA*: Invasion gene; *stn*: Enterotoxin gene; *hilA*: Hyper-invasive locus gene. **+**—positive result; “**−**”—negative result.

**Table 2 foods-11-02924-t002:** Number and (%) of antimicrobial resistant/susceptible *Salmonella* strains (*n* = 53) isolated from the investigated buffalo meat specimens.

Antimicrobial Agents	Susceptible	Intermediate	Resistant
E	0 (0%)	0 (0%)	53 (100%)
S	0 (0%)	1 (1.9%)	52 (98.1%)
CM	1 (1.9%)	2 (3.8%)	50 (94.3%)
CPM	7 (13.2%)	5 (9.4%)	41 (77.4%)
NA	15 (28.3%)	3 (5.7%)	35 (66.0%)
STX	21 (39.6%)	3 (5.7%)	29 (54.7%)
CET	24 (45.3%)	2 (3.8%)	27 (50.9%)
AMP	30 (56.6%)	1 (1.9%)	22 (41.5%)
TE	31 (58.5%)	4 (7.5%)	18 (34.0%)
ENR	37 (69.8%)	0 (0%)	16 (30.2%)
CIP	39 (73.6%)	3 (5.6%)	11 (20.8%)
MEM	44 (83.0%)	2 (3.8%)	7 (13.2%)
CTX	46 (86.8%)	2 (3.8%)	5 (9.4%)
GEN	49 (92.5%)	0 (0%)	4 (7.5%)
IPM	50 (94.3%)	1 (1.9%)	2 (3.8%)
AMK	52 (98.1%)	0 (0%)	1 (1.9%)

Legend: E: erythromycin; S: streptomycin; CM: clindamycin; CPM: cefepime; NA: nalidixic acid; SXT: trimethoprim/sulfamethoxazole; CET: cephalothin; AMP: ampicillin; TE: tetracycline; ENR: en-rofloxacin; CIP: ciprofloxacin; MEM: meropenem; CTX: cefotaxime; GEN: Gentamicin; IMP: imipenem; AMK: amikacin.

**Table 3 foods-11-02924-t003:** Antimicrobial resistance profile and MARs indexes of the isolated *Salmonella* serovars (*n* = 53) from buffalo meat.

*Salmonella* Strains (n = Number)	Antimicrobial Resistance Profile	MAR Index
*S*. Enteritidis	E, S, CM, CPM, NA, SXT, CET, AMP, TE, ENR, CIP, MEM, CTX, GEN, IPM, AMK	1
*S*. Enteritidis	E, S, CM, CPM, NA, SXT, CET, AMP, TE, ENR, CIP, MEM, CTX, GEN	0.875
*S*. Enteritidis	E, S, CM, CPM, NA, SXT, CET, AMP, TE, ENR	0.625
*S*. Enteritidis	E, S, CM, CPM, NA, SXT, CET, AMP, TE	0.563
*S*. Enteritidis (n = 2)	E, S, CM, CPM, NA, SXT, CET	0.438
*S*. Enteritidis	E, S, CM, CPM, NA, SXT	0.375
*S*. Enteritidis	E, S, CM, CPM, NA	0.312
*S*. Enteritidis	E, S, CM, CPM	0.250
*S*. Enteritidis (n = 2)	E, S, CM	0.187
*S*. Typhimurium	E, S, CM, CPM, NA, SXT, CET, AMP, TE, ENR, CIP, MEM, CTX, GEN, IPM	0.938
*S*. Typhimurium	E, S, CM, CPM, NA, SXT, CET, AMP, TE, ENR, CIP, MEM	0.750
*S*. Typhimurium	E, S, CM, CPM, NA, SXT, CET, AMP, TE, ENR	0.625
*S*. Typhimurium (n = 2)	E, S, CM, CPM, NA, SXT, CET, AMP	0.500
*S*. Typhimurium	E, S, CM, CPM, NA, SXT	0.375
*S*. Typhimurium	E, S, CM	0.187
*S*. Typhimurium	E, S	0.125
*S*. Montevideo	E, S, CM, CPM, NA, SXT, CET, AMP, TE, ENR, CIP, MEM, CTX, GEN	0.875
*S*. Montevideo	E, S, CM, CPM, NA, SXT, CET, AMP, TE, ENR	0.625
*S*. Montevideo	E, S, CM, CPM, NA, SXT, CET	0.438
*S*. Montevideo (n = 2)	E, S, CM, CPM, NA	0.312
*S*. Montevideo	E, S	0.125
*S*. Rissen	E, S, CM, CPM, NA, SXT, CET, AMP, TE, ENR, CIP, MEM, CTX	0.812
*S*. Rissen	E, S, CM, CPM, NA, SXT, CET, AMP, TE, ENR, CIP	0.687
*S*. Rissen	E, S, CM, CPM, NA, SXT, CET, AMP	0.500
*S*. Rissen	E, S, CM, CPM	0.250
*S*. Rissen	E, S, CM	0.187
*S*. Infantis	E, S, CM, CPM, NA, SXT, CET, AMP, TE, ENR, CIP, MEM	0.750
*S*. Infantis	E, S, CM, CPM, NA, SXT, CET, AMP, TE	0.563
*S*. Infantis	E, S, CM, CPM, NA, SXT, CET	0.438
*S*. Infantis	E, S, CM, CPM, NA	0.312
*S*. Infantis	E, S, CM, CPM	0.250
*S*. Virchow	E, S, CM, CPM, NA, SXT, CET, AMP, TE, ENR, CIP	0.687
*S*. Virchow	E, S, CM, CPM, NA, SXT, CET, AMP	0.500
*S*. Virchow	E, S, CM, CPM, NA	0.312
*S*. Virchow	E, S	0.125
*S*. Essen	E, S, CM, CPM, NA, SXT, CET, AMP, TE, ENR, CIP	0.687
*S*. Essen	E, S, CM, CPM, NA, SXT, CET	0.438
*S*. Essen	E, S, CM	0.187
*S*. Tsevie	E, S, CM, CPM, NA, SXT, CET, AMP, TE, ENR, CIP	0.687
*S*. Tsevie	E, S, CM, CPM, NA	0.312
*S*. Dublin	E, S, CM, CPM, NA, SXT, CET, AMP, TE, ENR	0.625
*S*. Dublin	E, S, CM, CPM	0.250
*S*. Anatum	E, S, CM, CPM, NA, SXT, CET, AMP, TE, ENR	0.625
*S*. Anatum	E, S	0.125
*S*. Derby	E, S, CM, CPM	0.250
*S*. Papuana	E, S, CM, CPM	0.250
*S*. Saintpaul	E, S, CM	0.187
*S*. Chester	E, S	0.125
*S*. Chester	E	0.0625
**n = 53**	Average MAR: 0.436	

E: erythromycin; S: streptomycin; CM: clindamycin; CPM: cefepime; NA: nalidixic acid; SXT: trimethoprim/sulfamethoxazole; CET: cephalothin; AMP: ampicillin; TE: tetracycline; ENR: enrofloxacin; CIP: ciprofloxacin; MEM: meropenem; CTX: cefotaxime; GEN: Gentamicin; IMP: imipenem; AMK: amikacin.

**Table 4 foods-11-02924-t004:** Classification of *Salmonella* isolates (n = 53) based on their antimicrobial resistance against the 16 tested drugs.

Antimicrobial Resistance Phenotype	Number and (%) of Isolates	MAR Index ^1^	Classification of Strains	
			Type of Resistance	Number and (%) of Isolates
E, S, CM, CPM, NA, SXT, CET, AMP, TE, ENR, CIP, MEM, CTX, GEN, IPM, AMK	1 (1.9%)	1	Pandrug-resistance	1 (1.9%)
E, S, CM, CPM, NA, SXT, CET, AMP, TE, ENR, CIP, MEM, CTX, GEN, IPM	1 (1.9%)	0.938	Extensively drug-resistant	4 (7.5%)
E, S, CM, CPM, NA, SXT, CET, AMP, TE, ENR, CIP, MEM, CTX, GEN	2 (3.8%)	0.875		
E, S, CM, CPM, NA, SXT, CET, AMP, TE, ENR, CIP, MEM, CTX	1 (1.9%)	0.8125		
E, S, CM, CPM, NA, SXT, CET, AMP, TE, ENR, CIP, MEM	2 (3.8%)	0.750	Multi-drug resistant	42 (79.2%)
E, S, CM, CPM, NA, SXT, CET, AMP, TE, ENR, CIP	4 (7.54%)	0.687		
E, S, CM, CPM, NA, SXT, CET, AMP, TE, ENR	5 (9.4%)	0.625		
E, S, CM, CPM, NA, SXT, CET, AMP, TE,	2 (3.8%)	0.562		
E, S, CM, CPM, NA, SXT, CET, AMP	4 (7.5%)	0.500		
E, S, CM, CPM, NA, SXT, CET	5 (9.4%)	0.437		
E, S, CM, CPM, NA, SXT	2 (3.8%)	0.375		
E, S, CM, CPM, NA	6 (11.3%)	0.312		
E, S, CM, CPM	6 (11.3%)	0.250		
E, S, CM	6 (11.3%)	0.187		
E, S	5 (9.4%)	0.125	Low-drug resistant	6 (11.3%)
E	1 (1.9%)	0.062		

^1^ MAR index: multiple antibiotic resistance index.

## Data Availability

Data are contained within the article.

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
