# Peer review of "Prevalence of Multidrug-Resistant Salmonella enterica Serovars in Buffalo Meat in Egypt"

_foods, 2022, doi:10.3390/foods11182924_

Round 1
Reviewer 1 Report (New Reviewer)
The manuscript is well written and has food safety/one-health implications. The language is easy to understand. Methodology, results, and discussion are well described with sufficient citations of appropriate references.
I have one suggestion as-Please in the introduction, provide the current buffalo census and carabeef production in Egypt to strengthen the hypothesis. Please also add the status of buffalo meat sector in Egypt whether based on spent animals, organized, etc.
Author Response
Reviewer #1
The manuscript is well written and has food safety/one-health implications. The language is easy to understand. Methodology, results, and discussion are well described with sufficient citations of appropriate references.
Our sincere thanks for taking the time to review this manuscript, and your close attention to detail. We highly appreciate your overall positive feed-back regarding the quality of the manuscript! Please see below for our responses to your comments:
I have one suggestion as - Please in the introduction, provide the current buffalo census and carabeef production in Egypt to strengthen the hypothesis. Please also add the status of buffalo meat sector in Egypt whether based on spent animals, organized, etc.
A paragraph (Lines 57-64) covering the livestock of buffalo as well as the importance of buffalo meat sector in Egypt is inserted in the "Introduction" section.
Thank you again!
Reviewer 2 Report (Previous Reviewer 2)
Dear Authors
my remarks have been met. Significant parts have been modified and the work is improved. I suggest only a minor revision of the English language because some sentences should be rephrased and also minor errors are scattered through the text
Author Response
Reviewer #2
Dear Authors,
my remarks have been met. Significant parts have been modified and the work is improved. I suggest only a minor revision of the English language because some sentences should be rephrased and also minor errors are scattered through the text.
The English language is edited and improved in the revised manuscript.
Our sincere thanks for taking the time to review this manuscript, and your close attention to detail. We highly appreciate your overall positive feed-back regarding the quality of the manuscript! The authors acknowledge the fact that the English content of the manuscript need some improvements. Accordingly, during the manuscript revision, each member of the research team tried to do her/his best to improve the English language. In addition, in the lack of “extensive English edits” raised concern by the reviewer, the authors believe that the improvement of the English language is solvable during the final “Pending English” status before publication, when the paper will be edited and finalized by the MDPI team. Thank you for your understanding and consideration!
Reviewer 3 Report (Previous Reviewer 3)
I proposed some corrections about this article in order to improve its quality. I see that authors responded to some of my previous comments. However authors state that Salmonella spp. was detected in 25 samples of tested buffalo meat. But the results concern all the colonies that were confirmed as Salmonella (n=53). In my opinion this might be due to duplication and testing of the same strains from the sample.
Additional comments;
Line 190 - Please refer to White-Kauffman le Minore scheme properly https://www.pasteur.fr/sites/default/files/veng_0.pdf see page 3: it should be White-Kauffmann-Le Minor scheme.
Lines 144-149 include test producers as in case of other media eg. XLD
Results
Lines 219-221 you declare that part on presumptive colonies is omitted, but not completely. You still indicate: “… presumptive isolates and revealed that only 53 (30.6%) of the 173 selected”
You have 25 samples positive for Salmonella and stick to that.
Moreover, instead of testing one positive colony per sample your serology results still refer to all presumptive colonies that were confirmed (53). Thus you just duplicate the results and instead 25 Salmonella strains you have 53.
I agree more than 25 strains is possible only on the condition that in some of the samples there were more than serovar identified. But you should justify in the manuscript why the study was done differently than ISO 6579-1:2017. According to ISO 6579-1:2017 you should test up to 5 presumptive colonies per sample. When the first colony is negative you test another up to 5. What were your reasons to do otherwise, please explain that and prove that the isolates you tested are not duplicates of the same strains.
Line 221 invA Please use italic for gene names
Lines 269-272 The sentence that you added completely undermines the ISO 6579-1:2017, norm which is used in official Salmonella research. Please rearrange it.
line 443 English correction "varirious"
Author Response
Reviewer #3
I proposed some corrections about this article in order to improve its quality. I see that authors responded to some of my previous comments. However, authors state that Salmonella spp. was detected in 25 samples of tested buffalo meat. But the results concern all the colonies that were confirmed as Salmonella (n=53). In my opinion this might be due to duplication and testing of the same strains from the sample.
Our sincere thanks for taking the time to review this manuscript, and your close attention to detail. We highly appreciate your overall positive feed-back regarding the quality of the manuscript! Please see below for our responses to your comments:
More than one colony is taken from the positive samples since it is possible to find more than one serovar in a single buffalo meat sample.
Additional comments;
Line 190 - Please refer to White-Kauffman le Minore scheme properly https://www.pasteur.fr/sites/default/files/veng_0.pdf see page3: it should be White-Kauffmann-Le Minor scheme.
Corrected in the revised manuscript.
Lines 144-149 include test producers as in case of other media eg. XLD.
Media used for the biochemical tests and their producers are inserted in the revised manuscript.
Results
Lines 219-221 you declare that part on presumptive colonies is omitted, but not completely. You still indicate: “… presumptive isolates and revealed that only 53 (30.6%) of the 173 selected”.
It is completely omitted in the revised manuscript and the sentence is edited.
You have 25 samples positive for Salmonella and stick to that. Moreover, instead of testing one positive colony per sample your serology results still refer to all presumptive colonies that were confirmed (53). Thus, you just duplicate the results and instead 25 Salmonella strains you have 53. I agree more than 25 strains is possible only on the condition that in some of the samples there were more than serovar identified. But you should justify in the manuscript why the study was done differently than ISO 6579-1:2017. According toISO 6579-1:2017 you should test up to 5 presumptive colonies per sample. When the first colony is negative you test another up to 5. What were your reasons to do otherwise, please explain that and prove that the isolates you tested are not duplicates of the same strains.
Besides the possibility of detection of more than Salmonella serovars in a single meat sample, more than one clone (isolate) of the same serovar (e.g. Salmonella Enteritidis) showed different virulence gene expression, as they show different antimicrobial resistance profile. Therefore, the count of the tested isolates was more than the number of positive samples.
Line 221 invA Please use italic for gene names.
It is carefully revised throughout the manuscript.
Lines 269-272 The sentence that you added completely undermines the ISO 6579-1:2017, norm which is used in official Salmonella research. Please rearrange it.
The sentence is deleted and the rest is edited.
line 443 English correction "varirious".
The correction is done.
Thank you again!
Round 2
Reviewer 3 Report (Previous Reviewer 3)
I see that authors responded to my comments. I believe the article is ready for publication. However, please consider adding a justification why you tested more than one positive isolate to the manuscript methods.
Author Response
Dear Reviewer,
According to your recommendation, we have inserted the following sentence in the Materials and methods section "
As has been previously indicated by several authors [3, 9, 10, 12, 16, 17], it is important to mention that more than one Salmonella isolate were tested from a single positive meat samples for the following reasons: (i) meat samples can contain more than one Salmonella serovars, (ii) two or more of the same Salmonella serovars can exhibit different antimicrobial resistant patterns, (iii) more than one isolates of the same or different Salmonella serovar taken from a single positive sample can exhibited different existence profile of the virulence genes tested."
Thank you again for your time and efforts!
This manuscript is a resubmission of an earlier submission. The following is a list of the peer review reports and author responses from that submission.
Round 1
Reviewer 1 Report
The article titled: Prevalence of multidrug-resistant Salmonella enterica serotypes isolated from buffalo meat in Egypt (Samir Mohammed Abd-Elghany*, Takwa Mohammed Fathy , Amira Ibrahim Zakaria , Kálmán Imre*, Adriana Morar , Viorel Herman , Raul PaÈ™calău , Laura Șmuleac , Doru Morar , Mirela Imre , Khalid Ibrahim Sallam) seems to be a very topical subject.
I have the following comments for the current version.
Major comments.
The most important omission is that lack of Salmonella serotyping according the Kauffmann–White classification. This should be done. The second major omission, which probably and unfortunately cannot be remedied, is the total lack of antimicrobial resistance genes.
Line 326 - The highest resistance rate of Salmonella isolates against E (100%) - its obvious! It has no effect on Enterobacteriaceae, like clindamycin. This should be corrected.
Table 3, 4 - deleted clindamycin, erythromycin
Specific comments:
The introduction is too general from the benefits of cattle, the value of beef and the general risks of meat from Salmonella spp. to diseases and symptoms of salmonellosis in humans. This is general information that can be shortened and expanded to include knowledge about Salmonella spp. in animals in Egypt, citing publications and reports on the occurrence of Salmonella in Egypt.
In the introduction, the literature cited comes from before 2013, and the only current literature is the self-quotes:
Sallam, K.I.; Abd-Elghany, S.M.; Imre, K.; Morar, A.; Herman, V.; Hussein, M.A.; Mahros, M.A. Ensuring safety and improving 427 keeping quality of meatballs by addition of sesame oil and sesamol as natural antimicrobial and antioxidant agents. Food Microbiol. 2021, 99, 103834. https://doi.org/10.1016/j.fm.2021.103834
Elshebrawy, H.A.; Mahros, M.A.; Abd-Elghany, S.M.; Elgazzar, M.M.; Hayashidani H., Sallam, K.I. Prevalence and molecular 435 characterization of multidrug-resistant and β-lactamase producing Salmonella enterica serovars isolated from duck, pigeon, and 436 quail carcasses in Mansoura, Egypt. LWT - Food Sci. Technol., 2021, 149, 111834. https://doi.org/10.1016/j.lwt.2021.111834
Line 124 ISO 6579:2002(en) Microbiology of food and animal feeding stuffs — Horizontal method for the detection of Salmonella spp. This standard has been withdrawn.
If the samples were actually taken in the period 2020-2021, then the applicable ISO 6579-1: 2017 standard should have been used, not the one from 20 years ago.
Authors should know how to proceed, in order to fully identify Salmonella spp. Both the serological and biochemical test results for a given bacterium should always be analyzed. The results do not include the antigenic patterns of the identified strains. Without serological testing, the biochemical and PCR tests themselves are inconsistent with the ISO 6579-1: 2017 standard and are not reliable.
Line 199-212 and Fig. 1A and Fig. 1B are unnecessary to include a bacterial strain as presumptive Salmonella - if the authors do not provide information about what other bacteria were - it does not bring anything new. This is simply ISO 6579-1: 2017 compliant as a step and can be safely ignored. There are known publications about difficulties in identifying Salmonella in meat also published in Foods: https: //doi.org/10.3390/foods10092177.
Line 141 - biochemical test were comercial or homemade?
Line 162, 165 - primers sequence should be added in methods.
Line 171 - 173 - description is too detailed
Line 173 - lack of name of molecular mass marker
Line 178 - both primers (forward and reverse) were using in one reaction?
Line 187 - breakpoint for celalotin according CLSI is only for MIC methods
Line 188 - breakpoint for clindamycin and enrofloxacin according CLSI is only for MIC methods
Line 192 - why E. coli ATCC 25922 but not Salmonella ATCC?
Line 200 - is repetition of line 138, 139 from methods. This should be corrected.
Line 207 - And the other strains were identified as?
Line 213-221 and line 226-230 are not the results of this study, therefore should be transferred to the Discussion and information on Salmonella spp. in meat in Egypt to the Introduction.
Figure 2 should be replaced by Table: The Salmonella enterica subsp. enterica variously identified serotypes isolated from samples of buffalo meat.
Line 251-254 it does not present the result, but a polemic that leads to wrong conclusions and should be removed. Moreover Real Time PCR based on detection of invA gene is more reliable, accurate, efficient, and faster than multiplex PCR, especially in meat samples.
I would like to see raw electrophoresis photo shown as Fig. 3. Could authors show native gel?
Fig 3 - Which type of UV Gel Documentation System was used?
Moreover C - : Control negative is E. coli, another negative control for reaction of PCR control should be water.
Lines 264-267 describe exactly what is shown in fig. 4, therefore fig. 4 should be deleted.
Lines 279-299 also do not present the result, they should be moved to the Discussion.
Lines 300-306 also do not present the result!
Lines 308-388 are also a mixture of Results and Discussions and duplicating information from the text in the tables.
The least effective antimicrobials against the tested Salmonella isolates were E , S, CM - it should be corrected. Erythromycin (line 326, 328) and clindamycin has no effect on Enterobacteriaceae. How it possible that 11% sensitive strain to clindamycin?
Table 2 -under table lack of antibiotic abbreviation
There is no Discussion section at all and it should be after the Results section. The conclusions are very general, it would be worth referring to the actual results from the current study.
Reviewer 2 Report
Dear Authors
It is an interesting report on Salmonella spp isolation from buffalo meat in Egypt also for its relevance for Public Health, both for zoonotic potential and antimicrobial resistance characteristics . However other authors have reported the presence of multiresistant Salmonella spp strains in Egypt from buffalo meat and the manuscripts may not be the first report. Moreover the discussion, that you put together with the results (I presume), is poor. You can find in attachment some remarks and suggestions that you should consider

Reviewer 3 Report
The authors present Salmonella occurrence in Buffalo meat in Egypt with particular emphasis on antimicrobial resistance of the pathogen. The subject addressed is important, especially that salmonellosis is one of the most important foodborne pathogens worldwide. Moreover, antimicrobial resistance is a problem of growing concern.
Anyhow, the article needs major revisions before possible publication. There is complete lack of "Discussion part" in the manuscript. Authors deliberate their findings combining it with presentation of results. However, in Microsoft Word template for the journal “Foods” Discussion should be separated from the results.
Moreover, extensive English correction by a professional or native English speaker is required.
Please see some additional comments:
Abstract
Line 35 Please change “strain” into “serovar”. I suggest to use the name serovar instead of serotype through the manuscript.
Introduction
Line 71. Please consider to begin the sentence not from “These bacteria…” but “The contamination might be caused by bacteria….”
Please note that you repeat the same information in Line 77 “…is reported worldwide as the most common causes of human gastroenteritis” and in line 81 “is recorded to be among the most common reasons of human gastroenteritis”
Line 105 “To date, no reports of Salmonella isolation from Egyptian buffalo meat…”
Are you sure? I found at least two publications on that (from 2014 and 2021)
Materials and Methods
Line 124 I would suggest to cite the current standard 6579 as the version from 2002 was already revised by ISO 6579-1:2017.
Line 142 Please note that TSI test is not only hydrogen sulfide but also glucose and lactose fermentation and gas production.
Line 145-146 Have you used White-Kauffman le Minore scheme for serotyping? If so please add this information.
Line 148-149 this sentence suggests that after biochemical and serological confirmation the strains were confirmed by PCR. No problem its up to you, however it would be surprising if the invA was negative after identifying the serovar. Have you had such cases?
If PCR was performed after biochemical confirmations. Transfer description of the serological identification after PCR.
Results
Lines 208-209 I think the information about presumptive colonies in results part is completely redundant. I suggests that this should be omitted also in the Fig 1.
Line 242 Please add space between “salmonellosisoutbreaks”
Line 244 - 245 I doubt that this is the first study. See: 10.21608/djvs.2021.94666.1048
Table 1. I would suggest to change „monitored virulence genes” into “tested virulence genes”
Lines 279-306 In my opinion considerations on the invA gene are redundant. Describing already obvious things known for years e.g. better specificity, sensitivity, and accuracy of invA PCR comparing with conventional methods that already been confirmed. Comparison of the results to other papers which “reported that all (100%) of the tested Salmonella isolates were positive for invA gene-specific for Salmonella spp.” is also useless in my opinion.
The authors themselves write that: “The invA gene is one of the most widely used genetic marker for the detection of Salmonella spp. in food” This is just the confirmatory method that was used. Separating the results from the discussion will help to avoid this.
Line 317-320 Please add some references that confirm this statement.
Lines 328-331 “…the management of Salmonella infections.” Do you mean in humans?
Please compile Table 3 and 4 into one table. Just add names of the serovars into Table 4 to avoid repetition of the same results. Please check the MAR indexes as I found some differences in both tables e.g. for E, S in on table 0.125 in the second 0.126
Please discuss resistance of particular Salmonella serovars as I think it should be included. The manuscript concerns resistance of Salmonella serovars and I think that is very important issue and should be deliberated. Have you identified any differences between serovars? How it looks in comparison to other parts of the world? You may focus at least at the serovars most commonly found in humans.